

# A model for French-press experiments of dry snow compaction

Colin R. Meyer[1], Kaitlin M. Keegan[2], Ian Baker[1], and Robert L. Hawley[3]

[1]Thayer School of Engineering, Dartmouth College, Hanover, NH 03755 USA
[2]Department of Geological Sciences and Engineering, University of Nevada, Reno, NV 89557 USA
[3]Department of Earth Science, Dartmouth College, Hanover, NH 03755 USA

*Correspondence to:* Colin R. Meyer (colin.r.meyer@dartmouth.edu)

**Abstract.** Compaction is the process by which snow densifies, storing water in alpine regions and transforming snow into ice on the surface of glaciers. Despite its importance in determining snow-water equivalent and glacier-induced sea level rise, we still lack a complete understanding of the physical mechanisms underlying snow compaction. In essence, compaction is a rheological process, where the rheology evolves with depth due to variation in temperature, pressure, humidity, meltwater. The
rheology of snow compaction can be determined in a few ways, for example, through empirical investigations (e.g. Herron & Langway,1980 *J. Glaciol.*), by microstructural considerations (e.g. Alley, 1987 *J. Phys.*), or by measuring the rheology directly, which is the approach we take here. Using a "French-press" compression stage, Wang and Baker (2013, *J. Geophys. Res.*) compressed numerous snow samples of different densities. Here we derive a mixture theory for compaction and air flow through the porous snow to compare against these experimental data. We find that a plastic compaction law explains
experimental results. Taking standard forms for the permeability and effective pressure as functions of the porosity, we show that this compaction mode persists for a range of densities and overburden loads. These findings suggest that measuring compaction in the lab is a promising direction for determining the rheology of snow though its many stages of densification.

## 1 Introduction

Snow densification in alpine and polar regions transforms snowflakes into ice crystals. On the surface of glaciers and ice sheets,
fresh snow is buried by new snow each winter, thereby slowly transforming into firn and then glacial ice as it compresses and descends. In cold and dry environments (e.g. melt-free accumulation areas of mountain glaciers, interior Greenland, and Antarctica), surface snow evolves due to temperature gradient metamorphism and atmospheric interactions (Dadic et al., 2008; Chen and Baker, 2010b). In the region below the surface, snow compaction is thought to occur in three stages (Herron and Langway, 1980; Arnaud et al., 2000; Cuffey and Paterson, 2010). In the first stage near the surface, compaction occurs by
grain growth due to sintering (Wilkinson, 1988) and rearrangement due to grain boundary sliding (Alley, 1987). In the second stage, the snow is old enough and far enough from the surface to be isothermal firn. Compaction in this stage is driven by the increasing overburden pressure inducing creep deformation (Wilkinson and Ashby, 1975; Wilkinson, 1988; Arnaud et al., 1998; Spencer et al., 2001). In the final stage, interconnected pores have closed-off and further compaction is caused by air bubble compression (Alley and Bentley, 1988; Salamatin et al., 1997; Gregory et al., 2014). In wet environments, such as
the percolation zone of mountain glaciers and Greenland as well as many ice shelves of Antarctica, compaction occurs by





a combination of dry snow compaction processes and refreezing of meltwater, which can either enhance or detract from the densification processes just mentioned (Colbeck, 1976; Machguth et al., 2016; Meyer and Hewitt, 2017). Although, meltwater percolation is an important part of the compaction process in many areas (e.g. Colbeck, 1972; Bartelt and Lehning, 2002; Wever et al., 2014; Steger et al., 2017), we will only consider dry snow compaction here.

An important reason for studying snow compaction on glaciers, ice sheets, and snowfields, is to connect a change in surface elevation to a volume of stored water. The total water volume stored in glaciers and snowpacks is important to know for current as well as future water resources and sea level rise considerations. Additionally, compaction is important for ice core analysis (Barnola et al., 1991; Goujon et al., 2003; Cuffey and Paterson, 2010). At the bottom of the firn column, the difference between the ice age and the gas age is an important input into paleoclimate reconstructions of temperature in ice cores

and must be estimated using a compaction model (Arnaud et al., 2000). Compaction can be measured through tracking relative displacements of metal markers (Hawley et al., 2004) or snow features in optical stratigraphy (Hawley and Waddington, 2011); autonomous phase-sensitive radio-echo sounding (Nicholls et al., 2015); vertical strain fiber optic sensors (Zumberge et al., 2002); the relative-displacement 'coffee can' technique (Hamilton et al., 1998; Hamilton and Whillans, 2000); and a continuously recording 'coffee can' method (Arthern et al., 2010; Stevens, 2018). The change in surface elevation $z_s$ due to

densification is a surface velocity $dz_s/dt = w_i(z_s)$, and the velocity within the snow induced by compaction varies with depth, i.e. $\partial w_i/\partial z \neq 0$. The body force acting on the snow is gravity, leading to an overburden pressure $\sigma_{\mathrm{o}} = \rho_s g z$ that also varies with depth. Given mass and momentum conservation, all that is needed to predict the densification rate (i.e. $w_i(z_s)$ and $\partial w_i/\partial z$) is a constitutive relationship between stress $\sigma_{\mathrm{o}}$ and strain rate $\partial w_i/\partial z$ in one vertical dimension. Thus, snow compaction can be thought of as a problem of rheology, where the rheology is complicated due to variation with depth, temperature, humidity,

water content, among many other physical processes. There are three common approaches to characterizing the rheology of a material (Tanner, 2000): empirically (e.g. Herron and Langway, 1980), from a micro-structural analysis (e.g. Alley, 1987), or experimentally. It is this last approach that we describe in this paper.

The standard compaction law applied to most glaciers and ice sheets is a one-dimensional relationship for the rate of change of density with depth, i.e.

$$\frac{\partial \rho_s}{\partial t} + w_i \frac{\partial \rho_s}{\partial z} = -\mathscr{C}\left(\rho_s, T, \dot{a}, \sigma_{\mathrm{o}}, \frac{\partial w_i}{\partial z}, \dots\right), \tag{1}$$

where $\mathscr{C}$ is the compaction function and can depend on the snow density $\rho_s$, temperature $T$, accumulation rate $\dot{a}$, overburden pressure $\sigma_{\mathrm{o}}$, vertical strain rate $\partial w_i/\partial z$, humidity, water saturation, grain size and potentially many other physical processes (Spiegelman, 1993; Arthern et al., 2010; Lundin et al., 2017). Herron and Langway (1980) take $\mathscr{C} = c(\dot{a}, T)\rho_s$, where $c$ is an empirical function, although, including the accumulation rate in the compaction function is dubious as it really should enter the

mass conservation equations as a boundary condition (Meyer and Hewitt, 2017). Other forms of the righthand side of equation (1) are discussed in Zwally and Li (2002); Reeh (2008); Morris and Wingham (2014); and Morris (2018). In this paper, we experimentally and mathematically analyze the function dependence of $\mathscr{C}$ on overburden pressure and strain rate.

We now summarize the outline of the paper. In section 2, we describe the laboratory compaction experiments of Wang and Baker (2013) and the samples used. In section 3, we construct a continuum model to describe the experiments and implement



| snow | density $\left(\mathrm{kg\ m^{-3}}\right)$ | specific surface area $\left(\mathrm{mm^{-1}}\right)$ |
|------|--------------------------------------------|------------------------------------------------------|
| SLT-1 | 322 | 26.8 |
| SLT-2 | 236 | 24.2 |
| SLT-3 | 233 | 23.0 |
| SLT-4 | 154 | 24.8 |

**Table 1.** Initial values of density and specific surface area for the four sintered low-temperature snow samples used in Wang and Baker (2013). Naming convention follows Fierz et al. (2009).

our model numerically. Lastly, in section 4, we compare the theoretical predictions for the snow compaction with the Wang and Baker (2013) data, showing that the theory does an excellent job explaining the snow compaction. We additionally discuss how this can be implemented into a compaction model such as equation (1) and our plans for future work, before a short conclusion section.

## 2 Experiments

To understand the microstructural origin of macroscopic snow material properties, the role of snow microstructure in avalanche initiation, and snow metamorphosis, Wang and Baker (2013) performed continuous compression tests on snow samples collected near Dartmouth College in Hanover, NH (USA). In their experiments, Wang and Baker (2013) focused on nine samples, ranging from freshly fallen, low-density snow that was relatively warm to snow that was collected during cold air temperatures (ca. -7°C to -9°C) and placed in a -30°C cold room for one year, allowing for the snow crystals to sinter. In the sintering process, bonds form between snow crystals (Male, 1980; Chen and Baker, 2010a; Wang and Baker, 2017) and the snow evolved into round and well-connected snow grains during the year in the cold room (Chen and Baker, 2010b). In this paper, we focus on the four sintered samples from Wang and Baker (2013), as they are most applicable to alpine snowpacks and firn on glaciers. The naming convention for these sintered low-temperature (SLT) samples follows Fierz et al. (2009), where the highest density sample is SLT-1 and the lowest density sample is SLT-4 (cf. table 1). Before compaction, the samples were vibrated to construct samples with different densities, thus, these samples represent a range of densities with relatively small differences in specific surface area (SSA; surface-to-volume ratio, viz. table 1). The snow samples were then extruded as cylinders, 15.7 mm in diameter and 18 mm tall, and placed on a Skyscan material testing compression stage (i.e. a french-press style coffee maker) inside of a microscopic computed tomography (micro CT) machine. A schematic of the compression stage is shown in figure 1. The samples were compressed at a constant rate of 12.7 mm/hour (i.e. 111 m/yr) for the full 5.7 mm dynamic range of the Skyscan compression stage. Due to termination effects, we only consider the first 5 mm of the range (Wang and Baker, 2013). The compression stage measured the load required to maintain a constant displacement rate for each snow sample. Wang and Baker (2013) then plot the loading stress as a function of the displacement, essentially a stress versus strain curve. In the next




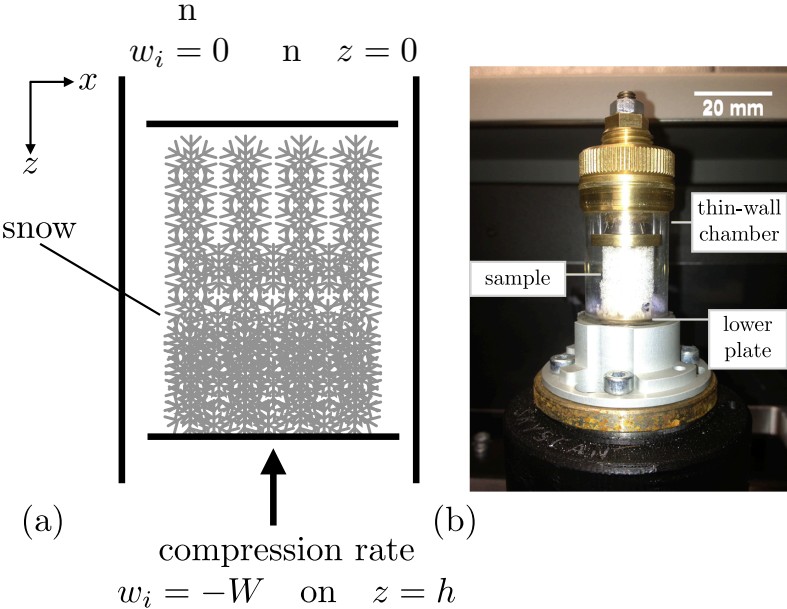

**Figure 1.** Schematic diagram of the snow french press: (a) idealized framework for the theory, including the boundary conditions and (b) experimental set-up from Wang and Baker (2013) showing the snow sample housed within the sample chamber and the upward motion of the lower plate. The sample is small enough that the effects of gravity can be ignored.

section, we develop a model for constant-displacement-rate compaction experiments and in section 4 we compare the model predictions for stress versus displacement against the Wang and Baker (2013) experimental data.

## 3 Model

Compaction occurs in a variety of natural and industrial processes, such as sedimentary basin formation, paper pulp dewatering, and particle flocculation, and has motivated numerous experiments and mathematical models (e.g. Landman et al., 1991; Fowler and Noon, 1999; Fowler, 2011; Hewitt et al., 2016a, b; Paterson et al., 2019). In this section, we describe the theory outlined by Hewitt et al. (2016b).

For dry snow that is composed of solely air and ice particles, the density of a volume of snow $\rho_s$ is given as

$$\rho_s = \phi\rho_a + (1 - \phi)\rho_i, \tag{2}$$

where $\phi$ is the porosity, i.e. the void space (Gray, 1996). The density of air is $\rho_a$ and the density of pure ice is $\rho_i$, both of which we treat as constant. From the expression in equation (2), it is clear that density variation in snow is due to an evolution of the porosity: near the surface of a glacier or snowpack, the snow air content is high and then density is closer to $\rho_a$, whereas at depth, the snow air content is low, and the density approaches $\rho_i$.



The evolution of porosity with depth is given by mass conservation, which for air and ice in snow can be written as

$$\frac{\partial (\rho_a \phi)}{\partial t} + \boldsymbol{\nabla} \cdot (\rho_a \phi \boldsymbol{u}_a) = m_s, \tag{3}$$

$$\frac{\partial [\rho_i (1-\phi)]}{\partial t} + \boldsymbol{\nabla} \cdot [\rho_i (1-\phi) \boldsymbol{u}_i] = -m_s, \tag{4}$$

where $\boldsymbol{u}_a$ is the air velocity, $\boldsymbol{u}_i$ is the ice velocity, and $m$ is the mass exchange between each phase due to sublimation. Again,

treating the densities of each substance as constant, neglecting the effect of sublimation, and restricting our attention to one vertical dimension, we can write this model as

$$\frac{\partial \phi}{\partial t} + \frac{\partial}{\partial z} (\phi w_a) = 0, \tag{5}$$

$$-\frac{\partial \phi}{\partial t} + \frac{\partial}{\partial z} [(1-\phi) w_i] = 0. \tag{6}$$

Now adding equations (5) and (6) gives the insight that decreasing the porosity requires squeezing the air out of the pore space,

or alternatively, that air motion leads to snow compaction, i.e.

$$\frac{\partial}{\partial z} [\phi w_a + (1-\phi) w_i] = 0. \tag{7}$$

In other words, there is an exact volumetric trade-off where the pore space occupied by air is filled by ice during compaction.

In the Wang and Baker (2013) experiments, the sample has an initial height of $z = h_0 = 18$ mm (cf. figure 1), where the sample is too small for gravitational effects to matter. Thus, we take the vertical coordinate system to increase downward. At

the top of the sample $z = 0$, the wall is impermeable, i.e. $w_a = w_i = 0$. Thus, we can integrate equation (7) to find that

$$w_i + \phi (w_a - w_i) = 0. \tag{8}$$

We model the air flow through porous snow using Darcy's law, which is given as

$$\phi (w_a - w_i) = -\frac{k(\phi)}{\mu} \frac{\partial p}{\partial z}, \tag{9}$$

for permeability $k(\phi)$, air viscosity $\mu$, and air pressure $p$. The remarkable thing about equation (8) is that this expression relates

the solid ice velocity $w_i$ to the Darcy velocity of the air flow through the pores, which reiterates the fact that mechanical compaction is facilitated by air flow. Thus, combining equations (8) and (9), we find that

$$w_i = \frac{k(\phi)}{\mu} \frac{\partial p}{\partial z}, \tag{10}$$

that is, the ice particle velocity is determined by the pressure gradient driving air motion.

Force balance in the vertical direction is given as

$$\frac{\partial \Sigma}{\partial z} = 0, \tag{11}$$

where $\Sigma$ is the vertical compression stress (Hewitt et al., 2016b). We define effective pressure $N$ as the difference between the compressive stress and air pressure, i.e. $N = \Sigma - p$, and write equation (10) as

$$w_i = -\frac{k(\phi)}{\mu} \frac{\partial N}{\partial z}. \tag{12}$$



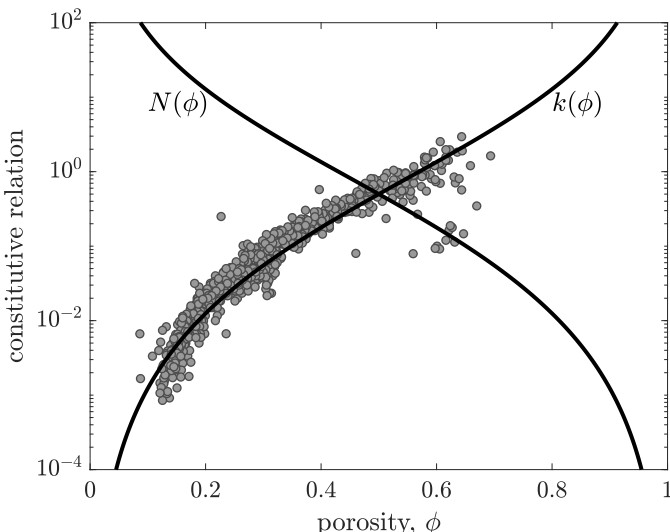

**Figure 2.** Effective pressure $N(\phi)$ and permeability $k(\phi)$ constitutive relations as functions of the porosity $\phi$ for the snow compaction theory. The grey dots are permeability measurements of firn cores from Summit, Greenland (Adolph and Albert, 2014) and show excellent agreement with the Carman-Kozeny permeability model.

As is typical in soil mechanics (Tulaczyk et al., 2000) and other applications of compaction (Fowler, 2011), we can relate the effective pressure $N$ to the porosity $\phi$ through an empirical constitutive relationship, so that $N(\phi)$. This relationship is a form of plasticity as there is a one-to-one correspondence between load and density without reference to displacement, strain rate, or stress history. Inserting this into equation (12) and putting that into equation (6), gives

$$5 \quad \frac{\partial \phi}{\partial t} = \frac{\partial}{\partial z}\left\{ (1-\phi)\left[-N'(\phi)\right]\frac{k(\phi)}{\mu}\frac{\partial \phi}{\partial z}\right\}, \tag{13}$$

which is a nonlinear diffusion equation for the porosity.

### 3.1 Constitutive relations

Equation (13) is a general model for mechanical compaction and the application to a specific system comes largely from the choice of constitutive relations for $k(\phi)$ and $N(\phi)$. A common choice for the permeability is the Carman–Kozeny model, i.e.

$$10 \quad k(\phi) = k_0 \frac{\phi^a}{(1-\phi)^b}, \tag{14}$$

which is commonly used throughout poromechanics (Rice and Cleary, 1976; McKenzie, 1984; Schoof and Hewitt, 2016) and has been extensively evaluated for snow and firn (Albert and Shultz, 2002; Adolph and Albert, 2013, 2014; Keegan et al., 2014), as shown in figure 2. Typical values for the exponents are $a = 3$ and $b = 2$. A competing model is a logarithmic permeability (Tait and Jaupart, 1992; Katz and Worster, 2008), of the form

$$15 \quad k_\ell(\phi) = -k_1 \phi^2 \ln(1-\phi), \tag{15}$$





and there are many other permeability models (e.g. Hewitt et al., 2016b). Here we use the Carman–Kozeny model. However, before future compaction experiments, we will measure the permeability and determine the constitutive relation as well as associated parameters that best represent each sample.

The dependence of effective pressure on the porosity of a sample has not to our knowledge been measured for snow or firn.

In other systems (e.g. Hewitt et al., 2016b), a plastic constitutive relationship between the effective pressure and porosity is given by

$$N(\phi) = N_0 \frac{(1-\phi)^n}{\phi^m}, \tag{16}$$

where typical exponents are $n = 3$ and $m = 2$ (cf. figure 2). An exponent of $n = 3/2$ in the numerator is sometimes used for very porous materials (Hewitt et al., 2016b). Just as for the permeability, in the future we plan to measure the constitutive

relation for effective pressure with porosity of each sample.

### 3.2 Boundary and initial conditions

As shown in figure 1, the samples are small and will be loaded from the bottom. The mathematical model for mechanical compaction results in a nonlinear diffusion equation for the porosity $\phi$, given in equation (13). Regardless of the choice of constitutive relations, two boundary conditions and one initial condition are required. At the top of the sample, as we mentioned

before, the ice and air flow must go to zero, therefore, the first boundary condition is

$$w_i = 0 \quad \text{at} \quad z = 0. \tag{17}$$

In terms of the porosity, this boundary condition translates into

$$\frac{\partial \phi}{\partial x} = 0 \quad \text{at} \quad z = 0, \tag{18}$$

which is a Neumann boundary condition.

At the top of the sample $z = h$, we apply a constant displacement rate, $W$, i.e.

$$w_i = W \quad \text{at} \quad z = h, \tag{19}$$

or equivalently

$$[-N'(\phi)] \frac{k(\phi)}{\mu} \frac{\partial \phi}{\partial z} = W \quad \text{at} \quad z = h, \tag{20}$$

which is also a Neumann boundary condition. Importantly, the constitutive equations play a role at the surface and not at depth.

Moreover, since the air pressure at the top of the sample is atmospheric, i.e. $p = 0$, the load applied to the sample to compress it at a constant displacement rate is given as

$$\Sigma = N(\phi) \quad \text{at} \quad z = h, \tag{21}$$

which we will compare to the Wang and Baker (2013) experimental data in the next section.





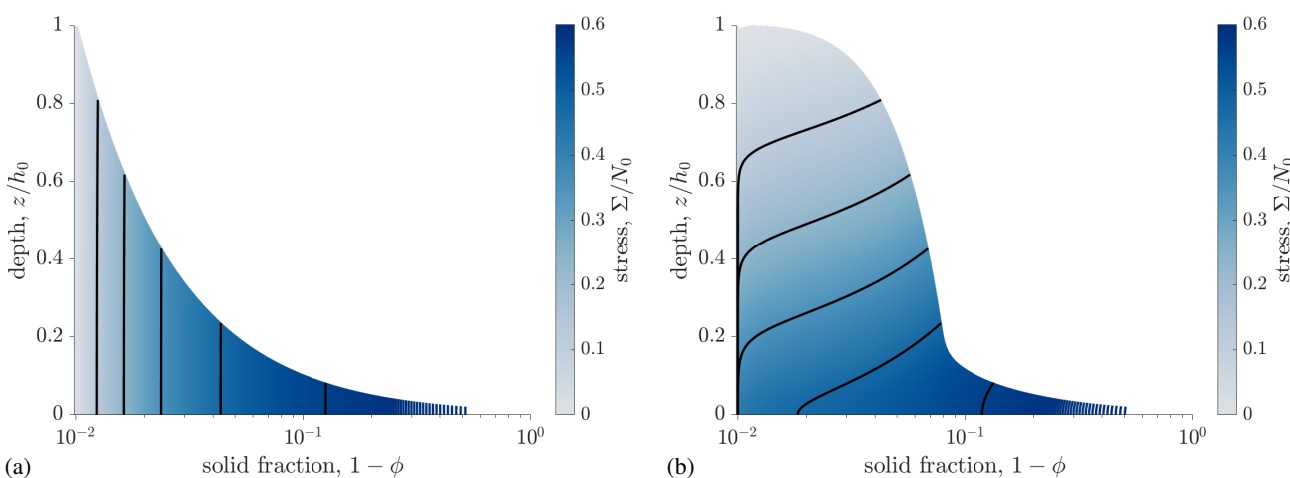

(a)      solid fraction, $1 - \phi$      (b)      solid fraction, $1 - \phi$

**Figure 3.** Compaction profiles for theoretical model showing solid fraction change ($1 - \phi$, where $\phi$ is the porosity) with depth (cf. Hewitt et al., 2016b). Colormap shows nondimensional stress and the black lines show the profiles at nondimensional time $t = 0.19, 0.38, 0.57, 0.76, 0.92$. Panels: (a) slow compaction ($\gamma = 1000$), where the the solid fraction is uniform with depth and (b) fast compaction ($\gamma = 1$), where significant compaction occurs near the lower part of the sample.

The initial condition is that the snow sample starts with a uniform porosity throughout, i.e.

$$\phi = \phi_0 \quad \text{at} \quad t = 0, \tag{22}$$

Additionally, the height of the snow sample evolves as a free-boundary during compaction according to

$$\frac{dh}{dt} = -W, \tag{23}$$

5    with the initial condition

$$h = h_0 \quad \text{at} \quad t = 0. \tag{24}$$

### 3.3 Nondimensional equations and numerical solution

In our theory for the snow compaction experiments of Wang and Baker (2013), we solve for the evolution of porosity with depth and time, as determined by equation (13) with the constitutive relations (14) and (16), the boundary conditions (18)

10    and (20), as well as the initial conditions (22) and (24). Taking the initial sample height $h_0$ to be a representative lengthscale, the displacement rate $W$ to be a characteristic velocity, as well as $k_0$ and $N_0$ to be scales for the permeability and effective pressure, respectively, we can nondimensionalize the variables as

$$z \rightarrow h_0 \hat{z}, \quad t \rightarrow \frac{h_0}{W} \hat{t}, \quad k \rightarrow k_0 \hat{k}, \quad N \rightarrow N_0 \hat{N},$$





where the hats represent nondimensionalization, although, for ease of notation, we immediately drop the hats. Inserting this nondimensionalization into the equations, we have

$$\text{(governing equation)} \quad \frac{\partial \phi}{\partial t} = \gamma \frac{\partial}{\partial z} \left\{ (1 - \phi) \left[ -N'(\phi) \right] k(\phi) \frac{\partial \phi}{\partial z} \right\}, \tag{25}$$

$$\text{(constitutive equations)} \quad N = \frac{(1 - \phi)^n}{\phi^m} \quad \text{and} \quad k(\phi) = \frac{\phi^a}{(1 - \phi)^b} \tag{26}$$

$$\text{(boundary conditions)} \quad \frac{\partial \phi}{\partial z} = 0 \quad \text{on} \quad z = 0; \quad \gamma \left[ -N'(\phi) \right] k(\phi) \frac{\partial \phi}{\partial z} = 1 \quad \text{and} \quad \frac{dh}{dt} = -1 \quad \text{on} \quad z = h, \tag{27}$$

$$\text{(initial conditions)} \quad \phi = \phi_0 \quad \text{and} \quad h = 1 \quad \text{at} \quad t = 0, \tag{28}$$

where

$$\gamma = \frac{k_0 N_0}{\mu h_0 W}, \tag{29}$$

is the ratio of pressure gradient $N_0/h_0$ to viscous resistance $\mu W/k_0$. Alternatively, this group can be thought of as the ratio of
effective pressure $N_0$ to viscous pressure $\mu W/h_0$ multiplied by the Darcy number $Da = k_0/h_0^2$ (Bear, 1972).

We solve equations (25)–(28) numerically using a finite volume discretization and the method of lines implemented in
Python (LeVeque, 2002). To facilitate the numerical computations, we map the compaction domain into a stationary domain
by using the change of variables $\xi = z/h$, which introduces a fictitious advection term into equation (25) (Hewitt et al., 2016b).
Solutions for two different values of $\gamma$ and the standard exponents (i.e. $a = 3$, $b = 2$, $n = 3$, $m = 2$) are shown in figure 3. The
two panels show the two primary regimes: for large values of $\gamma$ (left panel, $\gamma = 1000$; figure 3a), the dynamics are quasistatic
and the porosity $\phi$ (or solid fraction, $1 - \phi$) is constant with depth. For smaller values of $\gamma$ (right panel, $\gamma = 1$; figure 3b),
compression begins at the bottom of the sample and the top of the sample remains at the starting porosity. As time goes on the
sample is compressed and more of the sample compacts.

## 4 Results and discussion

Here we compare the predictions of the theory outlined in the previous section with the experimental data of Wang and Baker
(2013), which we described in the section 2. A key output of these experiments was the applied load required to compact
the snow as a function of displacement, during the constant displacement rate experiments. These data are shown in figure 4.
The lowest density snow SLT-1 withstands the least stress as a function of displacement. As the density of the snow samples
increases, so does the required stress for a given displacement. A similar observation can be made for the theory: the applied
load given by equation (21) and can be written dimensionally as

$$\Sigma = N_0 \frac{(1 - \phi)^n}{\phi^m} \quad \text{at} \quad z = h, \tag{30}$$

which states that the applied load $\Sigma$ increases as the porosity $\phi$ decreases (cf. figure 2). In other words, for a given displacement,
the stress in the theory will also increase as the initial density increases, which is also true for the experimental data. This makes
sense given that the compaction occurs through air evacuating the pore space. Thus, if there is less pore space to start with, a

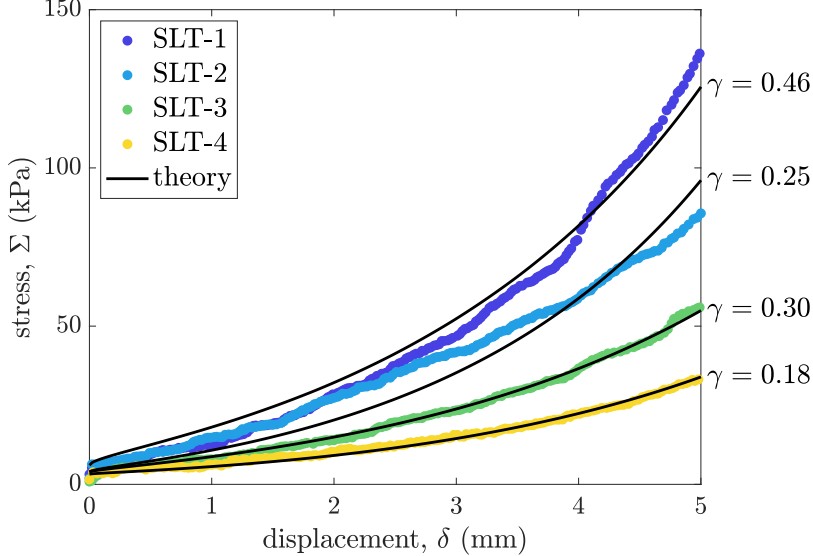

**Figure 4.** Comparison of theory with french press experiments from Wang and Baker (2013): four snow samples were compacted in a skyscan micro CT scanner at a constant displacement rate (cf. section 2 and figure 1). Using the theory outlined in section 3, the starting density of each sample, and reasonable parameters, we show agreement between theory and experimental data. For the theory, the constitutive exponents are the typical values ($a = 3$, $b = 2$, $m = 2$, $n = 2$), the stress was scaled (i.e. $N_0 = 30$ kPa, $\sigma_f = 3$ kPa), and the values of $\gamma$ for each curve are shown on the right.

great load is required to cause equivalent compaction. The theory for the load as a function of the displacement makes sense qualitatively, but we can qualitatively compare the theory to the experiments in the following way: we run the full model, equations (25)–(28), and evaluate the stress in equation (30) using the value of the porosity $\phi$ at the top of the sample $z = h$ and plot it against the displacement $\delta = h_0 - h$. The results for reasonable parameters are shown as black lines in figure 4.

The model, therefore, contains three inputs and seven parameters. The most important and most uncertain parameter is $\gamma$. Due to the fact that the permeability prefactor $k_0$ and the plastic effective pressure prefactor $N_0$ are unconstrained, we choose representative values for each sample. Although $k_0$ only appears within $\gamma$, the plasticity prefactor $N_0$ is required to scale the theory output to compare with experiments. The next four parameters are the exponents $a$, $b$, $m$, and $n$, which have expected values, but again have not been measured for these samples. Therefore, we only examine small departures from typical values.

Additionally, as we will see, there is a small amount of friction in the compression device and so a small constant stress $\sigma_f$ is added to the stress output from the theory, i.e. $\Sigma = N(\phi) + \sigma_f$ at $z = h$. This approach was also used by Hewitt et al. (2016b) in their experiments. The inputs are the initial porosity of the sample, which is set by the initial density measured by Wang and Baker (2013). Another input to the problem is the initial height of the sample, $h_0$. The total experiment run time $t_f$ is also an input and set by the final displacement divided by the displacement rate, i.e. $t_f = (h_0 - h_f)/W$, which is about 24 minutes.

The values of the fixed parameters used for the theory lines in figure 4 are the typical exponents (i.e. $a = 3$, $b = 2$, $m = 2$, $n = 2$) and the load scalings, $\sigma_f = 3$ kPa and $N_0 = 30$ kPa. The value of $\gamma$ for each curve is shown on the figure and is $\gamma = 0.46$



(SLT-1), $\gamma = 0.25$ (SLT-2), $\gamma = 0.29$ (SLT-3), and $\gamma = 0.18$ (SLT-4). In selecting these values, we successfully (1) found fixed load parameters that worked for all of the experiments and (2) kept the constitutive exponents as the typical values. Thus, $\gamma$ is the only parameter varies between theory predictions for each experiment, which is to be expected due to potential variation of $k_0$ with grain size. It is worth emphasizing that this is not a systemic nonlinear best fit analysis for seven parameters, rather

the agreement between theory and experiment demonstrates a physically motivated model with a single under-constrained parameter.

In general, there is excellent agreement between the theoretical predictions and the experimental data. For the bottom two samples, SLT-3 and SLT-4, the theory captures all of the major data trends and all falls well-within any experimental scatter. For the top sample SLT-1, the theory does a reasonable job capturing the data trend, yet for these parameters does not follow

the data points exactly. A better fit can be obtained by changing the parameters, indicating that this sample may be a different compaction regime than SLT-3 or SLT-4. However, the fit is reasonable enough that this sample is likely just on the edge of a new regime, if at all. In contrast to the other samples, the theory does not adequately capture the data trend of the middle sample, SLT-2. This sample is interesting because it has almost the same density and specific surface area as SLT-3, yet responded very differently to compression loading. Due to the small size of the samples, i.e. 15.7 mm in diameter and 18 mm tall, the likelihood

of defects or inhomogeneities dominating the results is quite high. Thus, the snow bonds in SLT-2 from prior sintering could have been particularly stubborn, requiring more load for a given displacement.

Another possibility for why the two snow samples SLT-1 and SLT-2 did not agree as well with the theory as SLT-3 and SLT-4, is that pressure sintering during the experiment allowed for greater bonding of snow crystals. Adding pressure is an efficient method of accelerating the rate of sintering and can lead to sinter rates that are orders of magnitude faster than by ambient

surface energy differences alone (Rahaman, 2007). For this reason, Wang and Baker (2013) attribute the increase in required load to accelerating the sintering and coarsening processes occurring within the snow samples. Willibald et al. (2019) also analyze sintering during compaction experiments and find that the sintering rate is enhanced. The plastic compaction theory we present in this paper does not include pressure processes and therefore, would not be able to describe the stress required to break sintered bonds, although this is a promising direction for future research.

The plastic compaction theory presented here can be related back to the general firn compaction model given in equation (1). Rearranging equation (6) gives

$$\frac{\partial \phi}{\partial t} + w_i \frac{\partial \phi}{\partial z} = (1 - \phi)\frac{\partial w_i}{\partial z}, \tag{31}$$

which is in the form of equation (1). Inserting Darcy's law (9) with mass conservation (5) and (6), the compaction function $\mathscr{C}$ is given by

$$\mathscr{C} = -\frac{1 - \phi}{\rho_a - \rho_i}\frac{\partial w_i}{\partial z} = \frac{1 - \phi}{\rho_a - \rho_i}\frac{\partial}{\partial z}\left[N'(\phi)\frac{k(\phi)}{\mu}\frac{\partial \phi}{\partial z}\right], \tag{32}$$

which shows the connection between compaction and air evacuating pore space as well as the role of the constitutive relations $N(\phi)$ and $k(\phi)$. In this way, measuring the parameters of these constitutive relations in the laboratory allows for predictions of compaction using $\mathscr{C}$ from equation (32).



For generalized viscous compaction, McKenzie (1984) starts from (31) and connects the divergence $\partial w_i/\partial z$ to an effective pressure $P_e$ through a compaction viscosity $\eta_c$, i.e.

$$\frac{\partial w_i}{\partial z} = \frac{P_e}{\eta_c},\tag{33}$$

which is the compaction law used in studies of temperate ice (Schoof and Hewitt, 2016; Hewitt and Schoof, 2017; Meyer et al.,
2018) and is equivalent to the viscous closure of a Röthlisberger channel (Nye, 1953; Fowler, 1984; Meyer et al., 2016, 2017). The McKenzie (1984) compaction law, equation (33), implies that

$$\mathscr{C} = -(1 - \phi)\frac{P_e}{\eta_c}.\tag{34}$$

Taking the effective pressure $P_e$ to be the overburden pressure, i.e. $P_e = \rho_s g z$, thereby setting the air pressure to zero, results in the pressure compaction model described by Cuffey and Paterson (2010) following Arthern et al. (2010). Although, this
compaction model is attractive because it connects viscous process to compaction, by setting the air pressure to zero (or even a constant), it fails to capture the evacuation of air from the pore space, which we have shown to be a very important process in the mechanical compaction of laboratory snow samples.

## 5  Conclusions

In this paper, we articulated a mathematical model to describe the snow compaction experiments of Wang and Baker (2013).
This model consists of mass and momentum conservation as well as porosity dependent permeability and plastic effective pressure constitutive equations. The outputs of the model are the snow density as a function of time and space as well as the stress as a function of displacement. Comparing the model outputs to the experimental data of Wang and Baker (2013) shows excellent agreement, especially for the low-density sintered snow. As the density increased, small discrepancies between model and theory emerged, potentially due to the necessity of creep or sample defects. Nevertheless, the excellent agreement between
theory and experiments suggests that measuring compaction in the lab is a promising direction forward for understanding snow compaction and that the plastic effective pressure as a function of porosity may be a key constitutive relation to quantify.

*Acknowledgements.* We wish to thank Alden Adolph and Xuan Wang for providing the data for figures 2 and 4, respectively, as well as Ian Hewitt, Harold Frost, and Yuan Li for insightful discussions.





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
