# Peer review of "A model for French-press experiments of dry snow compaction"

_The Cryosphere, 2019_

## Referee Comment (RC1) · Anonymous Referee #1 · 8 Dec 2019

This paper outlines a theoretical model to describe previous snow compaction experiments in a 'french-press'-style setup. The model describes unidirectional compaction of a plastic two-phase mixture, and is based on previous modelling derived for mixtures of deformable solids in a liquid suspension. The authors demonstrate that the model can give reasonable agreement with the previous experimental measurements, suggesting that this way of describing and snow-air mixture is plausible, and providing a possible means to calibrate material properties (compressive strength and permeability).

The paper is well written, clear and informative, and I think it should be published in this journal. I have only a few comments/questions that the authors could consider:

1) It would be interesting to report (and plot) how much differential compaction there

is in the samples for the experimental comparison in figure 4. The values quoted for gamma are fairly small, which suggests that there is quite a significant gradient of porosity (or effective stress/pressure) - i.e. the experiments were in a regime more like fig.3(b) than fig.3(a). Presumably Wang and Baker (2013) had some sense of whether they thought they were compressing in such a manner that the sample remained uniform in depth or not (?), and it would be interesting to see what the theory predicts. What would the large-gamma limit curves look like for the different experiments reported in figure 4, and how much above them is the load that was actually measured?

2) What is pore-scale Reynolds number of the air flow in the experiments? Is it always sufficiently low that Darcy's law can be reasonably assumed? One might also consider air compressibility - is it always reasonable to assume the density is constant in equation (3)?

3) At the end of the paper, the concept of viscous compaction is briefly introduced. This is interesting, and raises the question of how one would tell whether the snow compressed by Wang and Baker was behaving plastically or viscously. The authors state that the viscous theory can't capture the evacuation of air, but this isn't really fair - it would be perfectly possible to derive the same model (i.e. a two-phase model capturing the motion of the air) used by the authors but with a viscous closure law for the effective pressure in place of the plastic one in (16). (i.e. using (33), with N taking the place of $P_e$). I can't immediately see why this would not be able to capture the behaviour observed in the experiments. It seems important that there is some discussion of this.

Some typos:

Equation (3) and (4) have '$m_s$' in them, but '$m$' is reported in the following sentence. Equation (18) should have a '$z$' not an '$x$'. Superfluous comma after 'Although' on line 9 of page 12.

---

## Short Comment (SC1) · 12 Dec 2019

This study provides an insightful perspective on the densification of snow. The theoretical model is clearly explained. The model agrees well with laboratory experiments for snow samples in the density range of 150 to 325 kg mˆ-3. This is even more notable given that the parameters were set at their standard values (thus not calibrated against other data) and kept constant for the different experiments.

Interestingly, the authors point out the transition to a different compaction regime for the higher density samples. From a firn modelling perspective, it is important to separate the different stages of densification. Plastic compaction seem to describe accurately the densification in the low density ranges. Additional processes cause the compaction

rates to deviate from the theoretical predictions at higher densities. In order to reach physically based firn densification models, the community needs to (1) identify the different compaction regimes and (2) use the physically accurate governing processes for each regime. This study undoubtedly contributes to a better understanding of the early compaction.

Further research will be required to constrain the physics prevailing in other regimes and at the transitions between the different regimes. Also, the question of scaling will need to be addressed. Does such model-observations agreements can still hold for field samples of firn and snow which are more heterogeneous in their intrinsic properties.

I think that the following typos need to be corrected: Equation (18): changing x to z. On the line above Equation (19): the authors refer to the "top of the sample" whereas they give the boundary condition for the bottom of the sample. Equation (32): I think that the (rho_a - rho_i) term should be on the numerator and not on the denominator. Equation (33): I am not sure whether the authors forgot the (rho_a - rho_i) term or if it is implicitly included in the effective pressure term (in which case the authors should explicitly mention this for the sake of clarity).
* * *

---

## Referee Comment (RC2) · Anonymous Referee #2 · 25 Jan 2020

This paper applies a theory developed by Hewitt et al to describe the squeezing out of fluid from a sold matrix to a problem which is not exactly similar, namely the compression of dry snow. The difference lies in the fact that the fluid in Hewitt's analysis is incompressible but the moist air in snow is not. The authors need to convince the reader that Hewitt's theory can be applied. Given that it can, the parameters derived by fitting the theory to experimental data are a useful first step in deriving a densification law for snow. The paper would be greatly improved if the author's followed the nomenclature established by Hewitt, rather than making somewhat perverse changes which lead to confusion when the two papers are compared. Given that Hewitt has given a very clear and complete derivation of the theory I am not sure that it needs to be repeated in this paper. It might be be better to have more

discussion of the results of the modelling and their implications for snow densification in general. The paper is mostly well-written and the diagrams adequate but there are a host of minor errors and omissions which I have noted on the draft version attached. Given that these can be sorted out the paper will make an interesting addition to the literature on snow densification and is well worth publishing. Please also note the supplement to this comment: https://www.the-cryosphere-discuss.net/tc-2019-253/tc-2019-253-RC1- supplement.pdf Interactive comment on The Cryosphere Discuss., https://doi.org/10.5194/tc-2019-253, 2019.

Please also note the supplement to this comment:
https://www.the-cryosphere-discuss.net/tc-2019-253/tc-2019-253-RC2-supplement.pdf

**Supplement:**

[revised manuscript text omitted]

---

## Author Comment (AC1) · 22 Feb 2020

**Response to referees for 'A model for French-press experiments of dry snow compaction'**

Colin Meyer, Kaitlin M. Keegan, Ian Baker, and Robert L. Hawley

February 20, 2020

We wish to thank the referees for highlighting potentially confusing aspects of the paper and suggesting areas for improvement. Throughout, the original comments are in black text and our responses are colored in blue text.

**Editor comments**

The paper relies on previously published experimental data, and introduces a mechanical compaction model directly inspired from two-phases theories described in numerous previous studies. That being said, the application of this type of model to snow, and the comparisons with experiments, are original and promising. Notably, the results points out the importance of physical effects (air flow) that are neglected in most existing snow compaction models. The paper would benefit from a clearer emphasis on the main novelties of the approach.

The premises and physical ingredients of the compaction model are clearly explained. Yet, some key information relative to the experimental setup seem to be missing, such that the relevance and applicability of the model is difficult to assess. These concern notably air flow conditions at the top and bottom plates of the press. Regarding results, one would have expected to see more detailed analyses of model behaviour (influence of model parameters, evolution of air pressure, etc.) and more thorough comparisons with experiments (e.g., can the authors effectively prove that air flow plays a crucial role in the experiments?, why did they choose to consider a constant $N_0$, while this parameter is probably expected to vary with initial porosity?). From my point of view, the presented results are not necessarily sufficient, and appear too preliminary, to warrant publication. Being not a specialist of the topic, I will however rely on the assessment of the referees on that matter.

In agreement with our response to the referee comments, we have enhanced our description of the air flow conditions at the top and bottom of the plates, as shown in line 20 on page 3 and line 7 on page 8. The role of $N_0$ in our theory is indeed important and unfortunately unknown. The constitutive equation, however, includes the dependence of $N_0$ on porosity $\phi$, and therefore, should not depend on the initial porosity. Given that this is the first theoretical explanation for experimental snow compaction, we do not feel that our results are too preliminary, and the referees agree.

As rightly pointed out in the paper, the issue of modelling snow compaction is of high importance for numerous applications. Yet, for the model proposed by the authors to be considered as validated in a wide range of conditions, more thorough sensitivity analyses and comparisons against experimental results would need to be presented (cf previous point).

We agree that more sensitivity tests would be beneficial. We are currently working to produce more experimental datasets and measuring $N(\phi)$ – stay tuned!

Although the paper is clearly written and pleasant to read, the writing style would need to be made more 'scientific' in several instances. This includes the title, as "French-press"

does not appear to be a standard denomination for mechanical tests. Also, figure 3 (one of the two figures actually presenting results) is relatively difficult to understand and would need to be better explained.

We thank the editor for the kind assessment. The term 'French press' does not appear to be a universal term for 'cafetière à piston' and appeared to be a stumbling block for several readers. The use of the term in this context is not intended as a 'standard denomination for mechanical tests' rather as an analog to a kitchen appliance with the intent of demystifying the experiments and sparking good-natured enjoyment for pursuing scientific truth. We have added the name 'cafetière à piston' to the abstract and main text.

**Referee 1 comments**

This paper outlines a theoretical model to describe previous snow compaction experiments in a 'french-press'-style setup. The model describes unidirectional compaction of a plastic two-phase mixture, and is based on previous modelling derived for mixtures of deformable solids in a liquid suspension. The authors demonstrate that the model can give reasonable agreement with the previous experimental measurements, suggesting that this way of describing and snow-air mixture is plausible, and providing a possible means to calibrate material properties (compressive strength and permeability). The paper is well written, clear and informative, and I think it should be published in this journal. I have only a few comments/questions that the authors could consider:

1. It would be interesting to report (and plot) how much differential compaction there is in the samples for the experimental comparison in figure 4. The values quoted for gamma are fairly small, which suggests that there is quite a significant gradient of porosity (or effective stress/pressure) - i.e. the experiments were in a regime more like fig.3(b) than fig.3(a). Presumably Wang and Baker (2013) had some sense of whether they thought they were compressing in such a manner that the sample remained uniform in depth or not (?), and it would be interesting to see what the theory predicts. What would the large-gamma limit curves look like for the different experiments reported in figure 4, and how much above them is the load that was actually measured?

   This is an excellent point. Wang and Baker (2013) do not indicate the degree of differential compaction, but in our future experiments it is something that we will examine.

2. What is pore-scale Reynolds number of the air flow in the experiments? Is it always sufficiently low that Darcy?s law can be reasonably assumed? One might also consider air compressibility - is it always reasonable to assume the density is constant in equation (3)?

   We agree that the Reynolds number and compressibility are both potentially important physical effects. The second reviewer also stated that compressibility warranted a greater discussion. We now include a specific computation for the Reynolds number in line 21 on page 3 and address the compressibility of air in lines 11-13 on page 4.

3. At the end of the paper, the concept of viscous compaction is briefly introduced. This is interesting, and raises the question of how one would tell whether the snow compressed by Wang and Baker (2013) was behaving plastically or viscously. The authors state

that the viscous theory can't capture the evacuation of air, but this isn't really fair - it would be perfectly possible to derive the same model (i.e. a two-phase model capturing the motion of the air) used by the authors but with a viscous closure law for the effective pressure in place of the plastic one in (16). (i.e. using (33), with $N$ taking the place of $P_e$). I can't immediately see why this would not be able to capture the behaviour observed in the experiments. It seems important that there is some discussion of this.

This is a very fair assessment and something that the second reviewer also addressed. The McKenzie (1984) discussion in the original draft was intended to speak to this point. In the updated manuscript, we note that this model is not unique, but rather an end-member (i.e. plastic) version of a broader class of compaction laws that could likely describe the data equally well. Additionally, we are currently working on implementing a viscous version of this compaction law – stay tuned!

Some typos:

- Equation (3) and (4) have $m_s$ in them, but $m$ is reported in the following sentence.

- Equation (18) should have a $z$ not an $x$.

- Superfluous comma after 'Although' on line 9 of page 12.

All fixed. Thanks!

**Verjans comment**

This study provides an insightful perspective on the densification of snow. The theoretical model is clearly explained. The model agrees well with laboratory experiments for snow samples in the density range of 150 to 325 kg m$^{-3}$. This is even more notable given that the parameters were set at their standard values (thus not calibrated against other data) and kept constant for the different experiments.

Interestingly, the authors point out the transition to a different compaction regime for the higher density samples. From a firn modelling perspective, it is important to separate the different stages of densification. Plastic compaction seem to describe accurately the densification in the low density ranges. Additional processes cause the compaction rates to deviate from the theoretical predictions at higher densities. In order to reach physically based firn densification models, the community needs to (1) identify the different compaction regimes and (2) use the physically accurate governing processes for each regime. This study undoubtedly contributes to a better understanding of the early compaction.

Further research will be required to constrain the physics prevailing in other regimes and at the transitions between the different regimes. Also, the question of scaling will need to be addressed. Does such model-observations agreements can still hold for field samples of firn and snow which are more heterogeneous in their intrinsic properties.

I think that the following typos need to be corrected: Equation (18): changing $x$ to $z$. On the line above Equation (19): the authors refer to the "top of the sample" whereas they give the boundary condition for the bottom of the sample. Equation (32): I think that the $(\rho_a - \rho_i)$ term should be on the numerator and not on the denominator. Equation (33): I am not sure whether the authors forgot the $(\rho_a - \rho_i)$ term or if it is implicitly included in

the effective pressure term (in which case the authors should explicitly mention this for the sake of clarity).

Many thanks to Vincent Verjans for the constructive comments. We fixed the typos.

**Referee 2 comments**

This paper applies a theory developed by Hewitt et al. (2016) to describe the squeezing out of fluid from a sold matrix to a problem which is not exactly similar, namely the compression of dry snow. The difference lies in the fact that the fluid in Hewitt's analysis is incompressible but the moist air in snow is not. The authors need to convince the reader that Hewitt's theory can be applied. Given that it can, the parameters derived by fitting the theory to experimental data are a useful first step in deriving a densification law for snow. The paper would be greatly improved if the author's followed the nomenclature established by Hewitt, rather than making somewhat perverse changes which lead to confusion when the two papers are compared. Given that Hewitt has given a very clear and complete derivation of the theory I am not sure that it needs to be repeated in this paper. It might be be better to have more discussion of the results of the modelling and their implications for snow densification in general. The paper is mostly well-written and the diagrams adequate but there are a host of minor errors and omissions which I have noted on the draft version attached. Given that these can be sorted out the paper will make an interesting addition to the literature on snow densification and is well worth publishing.

Thanks to the second referee for a thorough read of our manuscript. Our analysis is indeed heavily influenced by the work of Hewitt et al. (2016), however, it is unfair to say that their work is not heavily inspired by earlier work by Landman et al. (1991) as well as Fowler and Noon (1999). On the topic of notation, we did not use the Hewitt et al. (2016) notation for the simple reason that we prefer $\phi$ for porosity and $N$ for effective pressure. Lastly, we have now added significant language describing that the air is free to flow out of the sample and is therefore incompressible. For example, lines 19-20 on page 3.

**Supplemental comments**

The referee also included comments directly on the pdf document. We made the many changes they suggest and thank them for helping us improve our paper. Additionally, we respond to a selection of these comments here:

- "but it is a proxy for overburden, which can quite reasonably be included"

  This is a fair point. However, the rate of accumulation should really affect the vertical velocity rather than the overburden pressure, thus, is better suited as a boundary condition, given snowfall also occurs at the surface boundary.

- "The crucial question here is whether the moist air can escape from the cylinder (i.e. whether the "plunger" is porous as in a cafetiere) or whether the gas is compressed as the volume decreases. By going to the Hewitt paper it is possible to deduce that the piston is porous... but that needs to be stated explicitly here."
  We have added text stating that the air can escape from the sample and should therefore be incompressible, e.g. lines 19-20 on page 3.

- "But if you tell the reader you are following Hewitt you should warn him that you use phi for porosity and Hewitt uses phi for the solid fraction. Otherwise it is very

confusing to compare the two expositions."
We now explicitly state that we use $\phi$ for porosity instead of solid fraction: lines 7 and 8 on page 4.

- "Hewitt explains pretty clearly that all that is happening here is that momentum transfer between fluid and matrix is included in the Darcy expression - so the equation is not 'remarkable' surely?"
  This is a fair criticism. We now removed this language.

- "with what parameters?"
  here we use the Cozeny-Karman with $a = 3$ and $b = 2$, which we now explicitly state.

- "$t$ has been used for dimensional time. You need to use different symbol"
  we changed this to $t/t_0$.

- "$N_0$ does appear in gamma according to eq. (29)"
  we agree and now add a statement to this effect.

- "I'm not sure about this argument. There are two different assumptions about the strength of the ice matrix - (1) your model that the effective yield stress is related to porosity and (2) the alternative idea that the flow is viscous. The question of the effect of air flow on the matrix could be included in a viscous flow type model I would suppose."
  We now address this point explicitly at the end of the discussion.

- "You seem to be dodging the big question - how does your compaction model compare to existing models applied to the same data?"
  This is an excellent lead-in to our future work!

**References**

A. C. Fowler and C. G. Noon. Mathematical models of compaction, consolidation and regional groundwater flow. *Geophys. J. Int.*, 136(1):251–260, 1999. doi: 10.1046/j.1365-246X.1999.00717.x.

D. R. Hewitt, D. T. Paterson, N. J. Balmforth, and D. M. Martinez. Dewatering of fibre suspensions by pressure filtration. *Phys. Fluids*, 28(063304):1–23, 2016. doi: 10.1063/1.4952582.

K. A. Landman, C. Sirakoff, and L. R. White. Dewatering of flocculated suspensions by pressure filtration. *Phys. Fluids*, 3(6):1495–1509, 1991. doi: 10.1063/1.857986.

D. McKenzie. The generation and compaction of partially molten rock. *J. Petrol.*, 25(3): 713–765, 1984. doi: 10.1093/petrology/25.3.713.

X. Wang and I. Baker. Observation of the microstructural evolution of snow under uniaxial compression using X-ray computed microtomography. *J. Geophys. Res.*, 118(A12):371–382, 2013. doi: 10.1002/2013JD020352.